Corrected: Publisher correction

# Fuelling conditions at staging sites can mitigate Arctic warming effects in a migratory bird

Eldar Rakhimberdiev [1,2], Sjoerd Duijns [1,3], Julia Karagicheva[1], Cornelis J. Camphuysen[1], VRS Castricum[#],
Anne Dekinga[1], Rob Dekker[1], Anatoly Gavrilov[4], Job ten Horn[1], Joop Jukema[5], Anatoly Saveliev [6],
Mikhail Soloviev [2,4], T. Lee Tibbitts[7], Jan A. van Gils [1] & Theunis Piersma[1,8]

Under climate warming, migratory birds should align reproduction dates with advancing plant and arthropod phenology. To arrive on the breeding grounds earlier, migrants may speed up spring migration by curtailing the time spent *en route*, possibly at the cost of decreased survival rates. Based on a decades-long series of observations along an entire flyway, we show that when refuelling time is limited, variation in food abundance in the spring staging area affects fitness. Bar-tailed godwits migrating from West Africa to the Siberian Arctic reduce refuelling time at their European staging site and thus maintain a close match between breeding and tundra phenology. Annual survival probability decreases with shorter refuelling times, but correlates positively with refuelling rate, which in turn is correlated with food abundance in the staging area. This chain of effects implies that conditions in the temperate zone determine the ability of godwits to cope with climate-related changes in the Arctic.

[1] NIOZ Royal Netherlands Institute for Sea Research, Department of Coastal Systems and Utrecht University, PO Box 59, 1790 AB Den Burg, Texel, The Netherlands. [2] Department of Vertebrate Zoology, Biological Faculty, Lomonosov Moscow State University, 119991 Moscow, Russia. [3] Department of Biology, Carleton University, 1125 Colonel By Drive, K1S 5B6 Ottawa, ON, Canada. [4] Directorate of Taimyrsky Reserves, 663305 Norilsk, Russia. [5] Haerdawei 62, 8854 AC Oosterbierum, The Netherlands. [6] Institute of Environmental Sciences, Kazan Federal University, 420097 Kazan, Russia. [7] U.S. Geological Survey Alaska Science Center, 4210 University Drive, Anchorage, AK 99508, USA. [8] Chair in Global Flyway Ecology, Conservation Ecology Group, Groningen Institute for Evolutionary Life Sciences (GELIFES), University of Groningen, PO Box 11103, 9700 CC Groningen, The Netherlands. Correspondence and requests for materials should be addressed to E.R. (email: eldarrak@gmail.com) or to T.P. (email: theunis.piersma@nioz.nl)
[#] A full list of consortium members appears at the end of the paper.

Global warming is not globally uniform. In the Arctic, climatic changes are the strongest[1,2], with the highest rates of advance in spring phenology, which contrasts with the slower changes in equatorial regions. Migratory bird populations that fail to maintain a match between the timing of breeding and the local phenology of resources have low reproductive output and show declines[3,4]. Some populations appear capable of tracking spring at the breeding grounds, but little is known about the fitness costs that such adjustments imply. We may expect these costs to be especially high in long-distance migrants whose annual cycles include use of widely separated places with different rates of phenological drift.

To examine the fitness trade-off involved in the adjustment of reproductive timing by Arctic breeding birds to advancements in the general phenology of the tundra, we studied bar-tailed godwits (Limosa lapponica taymyrensis; hereafter godwits). This population is among the several long-distance migratory shorebirds that travel from wintering grounds in West Africa to breed in the Russian Arctic with a single refuelling stop (of ca. 25 days) in the Wadden Sea of north-western Europe[5,6]. Godwits arrive in the intertidal areas of the Wadden Sea in late April—early May after a non-stop flight of about 5000 km, and spend most of their available time feeding[7,8]. The birds double their body mass in less than a month, which fuels their next 5000 km long migratory flight to the Arctic breeding grounds[9].

Our study combines data on godwits and their food resources at the main wintering, spring refuelling, and breeding sites along the flyway. To track in detail how individual birds connect these sites, we instrumented eight godwits with satellite transmitters in 2016. To estimate population trends, we counted birds each winter from 2002 to 2016 at the main wintering area, the Banc d'Arguin, Mauritania, West Africa. For godwits staging in the Wadden Sea during northward migration, we assessed the relationship between the annual refuelling rates and the density of their main prey, adult lugworms (Arenicola marina), using a 21-year dataset and a hierarchical Bayesian model that accounted for year-specific arrival dates and arrival mass of godwits.

Godwits are sexually dimorphic, with females being 17% larger than males[6]. For this reason, we modelled arrival mass and refuelling rates at the Wadden Sea as being sex dependent. The departure dates from the Wadden Sea were derived by subtracting duration of migration (obtained in 2016 with satellite transmitters) from the observed arrival dates on the breeding grounds at the Taimyr Peninsula, Russian Arctic (Supplementary Table 1). For each sex we then estimated the effects of refuelling time and refuelling rate on subsequent survival probability of individually colour-marked birds. To evaluate the effects of climate change on the breeding ecology of godwits, we assessed the influence of dates of snowmelt[10] and emergence of the main food of shorebird chicks (adult crane flies; Tipula sp.[11,12]) on arrival and breeding dates of godwits over a 25-year period (1992–2016).

The emergence of crane flies in the Arctic is advancing in concert with advancements of snowmelt. The godwits follow these changes and arrive on the breeding grounds earlier and initiate nests earlier too. They manage this by shortening the time for refuelling in the Wadden Sea, but this comes at the cost of lower subsequent survival probability as the shorter time is not fully compensated by higher refuelling rates. To some extent higher food densities in the Wadden Sea would increase the level at which Arctic warming can be mitigated on temperate shores.

## Results

### Spring migration and breeding phenology of godwits. The date of snowmelt at the godwit breeding grounds on Taimyr has shifted forward by $0.73 \pm 0.16$ s.e.m. d year$^{-1}$ ($P < 0.001$, Table 1,

row 1, Fig. 1a). The main prey for godwit chicks, the adult Tipula crane fly, has responded to changes in snowmelt by advancing its first emergence time by $0.38 \pm 0.14$ ($P = 0.018$) days per day of advancing snowmelt (Table 1, row 2, Fig. 1b). The godwits advanced their arrival on the breeding grounds by $0.22 \pm 0.071$ days ($P = 0.006$, Table 1, row 3) and advanced clutch initiation by $0.56 \pm 0.17$ days ($P = 0.006$, Table 1, row 4) per day of snowmelt advancement. The path analysis of the statistical causality in the phenological data (Fig. 1b and Supplementary Table 2), revealed that breeding date of godwits was driven by arrival date, and not by the timing of crane fly appearance, while godwit arrival on the breeding grounds, as well as crane fly appearance, was driven by timing of snowmelt.

Satellite tracking of godwits confirmed the previously suggested migration strategy[6] of swift and direct migrations between West Africa and the Wadden Sea and between the Wadden Sea and Taimyr (Fig. 1c). Individual birds stopped only on six occasions during ten migration legs (two from West Africa to Wadden Sea and eight from Wadden Sea to Taimyr) for a mean period of 1.8 d (maximum 2.8 d) and completed the 5000 km migration leg between the Wadden Sea and Taimyr in ca. 5.5 days (mean = 5.16, s.d. = 2.09, median = 5.72, $n = 8$). Date of arrival at the Wadden Sea refuelling site did not significantly change over time ($-0.04 \pm 0.06$ d year$^{-1}$, $P = 0.49$, Table 1, row 5, Fig. 1a). However, dates of arrival on the breeding grounds and of clutch initiation advanced ($-0.70 \pm 0.27$ d year$^{-1}$, $P = 0.03$, Table 1, row 6; and $-0.28 \pm 0.10$ d year$^{-1}$, $P = 0.01$, Table 1, row 8). This advancement was achieved by the birds showing a tendency to shorten their refuelling time in the Wadden Sea by 16% between 1995 and 2015 ($-0.24 \pm 0.13$ d year$^{-1}$, $P = 0.08$, Table 1, row 9).

### Annual survival probability of godwits. The fuel load accumulated in the Wadden Sea is the product of refuelling time and refuelling rate[13]. This relationship means that birds can reach the same level of body stores in a shorter time by increasing their refuelling rates. However, in association with the reduction in refuelling time, during the study period annual survival probability of godwits decreased by 2% ($-0.08 \pm 0.01$ on logit scale, $\Delta$QAICc = 35.11, Table 1, row 10). This negative effect of the reduced refuelling time on survival was stronger in females ($2.86 \pm 0.43$, Fig. 2b) than in males ($1.43 \pm 0.44$, Fig. 2c), and the difference between the sexes was statistically significant ($\Delta$QAICc = 4.73 Table 1, row 12). The refuelling rate effect on survival ($0.96 \pm 0.45$ on logit scale, $\Delta$QAICc = 2.14, Table 1, row 13) did not significantly differ between sexes ($\Delta$QAICc = 1.61, Table 1, row 14).

Refuelling rates correlated with densities of adult lugworms ($0.20 \pm 0.03$, $P < 0.0001$, Table 1, row 15, Fig. 2e and g). Even though lugworm densities did not increase statistically significantly during the study period ($0.15 \pm 0.14$, $P = 0.29$, Table 1, row 17), the refuelling rates of godwits did ($0.07 \pm 0.02$ g d$^{-1}$ year$^{-1}$ in both sexes, $P = 0.006$, Table 1, row 18, Fig. 2d and f). The increased refuelling rate partly offset the effect of reduced refuelling time on survival probability. In males the annual survival probability in 2015 was 3% higher than expected if refuelling rates would not have changed since 1995. In females, the compensation in survival was 2%. The decrease in annual survival probability caused by changes during refuelling in the Wadden Sea would thus have contributed to the $4.0 \pm 1.6$% per year population decline ($P(\lambda < 1) = 0.994$, Table 1, row 20) revealed by midwinter counts on the Mauritanian wintering grounds (Fig. 1d).

Thus, to maintain the annual survival probability as initially observed, godwits would need to refuel at higher rates than

**Table 1 Details on results of the analysis**

| Statements | Supporting test details | Test results |
|---|---|---|
| 1. Snowmelt dates on Taimyr advanced over years | Comparison of models with and without time trend in the snowmelt dates | Slope = −0.73 ± 0.16, $N$ = 24, d.f. = 1, $F$ = 22.31, $P$ < 0.001 |
| 2. Crane fly emergence dates correlated with snowmelt dates | Comparison of models with and without effect of snowmelt on crane fly emergence dates | Slope = 0.38 ± 0.14, $N$ = 16, d.f. = 1, $F$ = 2.69, $P$ = 0.02 |
| 3. Time of arrival to Taimyr correlated with snowmelt dates | Comparison of models with and without effect of snowmelt on time of arrival to Taimyr | Slope = 0.22 ± 0.07, $N$ = 21, d.f. = 1, $F$ = 9.64, $P$ = 0.006 |
| 4. Breeding dates correlated with snowmelt dates | Comparison of models with and without effect of snowmelt on breeding dates | Slope = 0.56 ± 0.17, $N$ = 13, d.f. = 1, $F$ = 11.39, $P$ = 0.006 |
| 5. Time of arrival to the Wadden Sea did not change over years | Comparison of models with and without time trend in mean date of arrival to Wadden Sea | Slope = −0.04 ± 0.06, $N$ = 24, d.f. = 1, $F$ = 0.49, $P$ = 0.49 |
| 6. Breeding dates advanced over years | Comparison of models with and without time trend in breeding dates | Slope = −0.70 ± 0.27, $N$ = 13, d.f. = 1, $F$ = 6.76, $P$ = 0.03 |
| 7. Crane fly emergence dates had tendency to advance over years | Comparison of models with and without time trend in crane fly emergence dates | Slope = −0.40 ± 0.21, $N$ = 16, d.f. = 1, $F$ = 3.57, $P$ = 0.08 |
| 8. Time of arrival to Taimyr advanced over years | Comparison of models with and without time trend in dates of arrival to Taimyr | Slope = −0.28 ± 0.10, $N$ = 21, d.f. = 1, $F$ = 7.32, $P$ = 0.01 |
| 9. Refuelling time tended to decrease over years | Comparison of models with and without time trend in the refuelling time | Slope = −0.24 ± 0.13, $N$ = 21, d. = 1, $F$ = 3.45, $P$ = 0.08 |
| 10. There is temporal trend in annual survival | Comparison of capture-recapture model with time trend in survival vs. model without time trend | Slope = −0.08 ± 0.01, $N$ = 3995, d.f. = 1, $\Delta QAIC_c$ = 35.11 |
| 11. There is no sex-specific difference in temporal trend in survival | Comparison of capture-recapture model with interaction between sex and time trend vs model without interaction | $Slope_{females}$ = −0.09 ± 0.02, $Slope_{males}$ = −0.07 ± 0.02, $N$ = 3995, d.f. = 1, $\Delta QAIC_c$ = 0.28 |
| 12. There is a difference between sexes in response of annual survival ($\Phi$) to changes in refuelling time | Comparison of capture-recapture model with interaction between sex and log(refuelling time) vs model without interaction | $Slope_{females}$ = 2.86 ± 0.43, $Slope_{males}$ = 1.43 ± 0.44, $N$ = 3995, d.f. = 1, $\Delta QAIC_c$ = 4.73 |
| 13. Annual survival ($\Phi$) is affected by refuelling rate | Comparison of capture-recapture model with and without log(refuelling rate) | Slope = 0.98 ± 0.46, $N$ = 3995, d.f. = 1, $\Delta QAIC_c$ = 2.03 |
| 14. There is no difference between sexes in response of annual survival ($\Phi$) to changes in refuelling rate | Comparison of capture-recapture model with interaction between sex and log(refuelling rate) vs model without interaction | $Slope_{females}$ = 1.14 ± 0.55, $Slope_{males}$ = 0.57 ± 0.75, $N$ = 3995, d.f. = 1, $\Delta QAIC_c$ = 1.58 |
| 15. Refuelling rates correlated with lugworm abundance | Comparison of models with and without lugworm abundance effect on mean sex-specific refuelling rates | Slope = 0.20 ± 0.03, $N$ = 40, d.f. = 1, $F$ = 49, $P$ < 0.0001 |
| 16. There was no statistically significant sex-specific difference in effect of lugworm density on refuelling rates | Comparison of model with multiplicative effect of sex and lugworm abundance on mean sex-specific refuelling rates with model additive model | $Slope_{females}$ = 0.21 ± 0.04, $Slope_{males}$ = 0.18 ± 0.04, $N$ = 40, d.f. = 1, $F$ = 0.18, $P$ = 0.67 |
| 17. Lugworm density did not change over years | Comparison of linear models of lugworm density with and without time trend | Slope = 0.15 ± 0.14, $N$ = 40, d.f. = 1, $F$ = 1.20, $P$ = 0.29 |
| 18. Refuelling rates have increased over years | Comparison of models with and without time trend in mean sex-specific refuelling rates | Slope = 0.07 ± 0.02, $N$ = 42, d.f. = 1, $F$ = 8.37, $P$ = 0.006 |
| 19. There was no sex-dependent difference in trend of refuelling rates over years | Comparison of model with multiplicative effect of sex and time on mean sex-specific refuelling rate with model additive model | $Slope_{females}$ = 0.07 ± 0.03, $Slope_{males}$ = 0.06 ± 0.03, $N$ = 42, d.f. = 1, $F$ = 0.05, $P$ = 0.84 |
| 20. There is a decline in population size over years | Estimation of proportion of growth rate $\lambda$ in MCMC samples being below 1 | $\lambda$ = 0.96 ± 0.016, $n_{years}$ = 15, $n_{counts}$ = 88, $P(\lambda < 1)$ = 0.994 |

The result-statements are presented in the order to which they are introduced in the narrative (note that some statements are implicit and do not show up in the text). $\Delta QAIC_c$ values were calculated as the 'simpler model'–'more complex model' (so that positive values mean that the more complex model is better). All effects of covariates on annual survival probability are estimated and presented on logit scale

measured in our study, and therefore would require even higher densities of lugworms. For full compensation (i.e. a refuelling rate of 6.6 g d$^{-1}$), males would need an average density of 20 lugworms m$^{-2}$. These densities occur in the Wadden Sea occasionally. Females, however, for full compensation would need to refuel at 9.9 g d$^{-1}$, which would require 32.7 lugworms m$^{-2}$, an average density never encountered by our monitoring effort in the Wadden Sea during the study period (Fig. 2h).

## Discussion

Our long-term, hemispheric scale observations suggest an important and previously unrecognized mechanism by which migratory birds cope with global change. Rather than the use of multiple sites simply being a liability[3,14], it may provide opportunities for among-season compensation[15]. In contrast to many other species[16–18], bar-tailed godwits adjusted their arrival on the breeding grounds and the onset of breeding, thereby tracking the seasonal advancement of their main arthropod prey on the breeding grounds. They achieved this by shortening their refuelling period in the Wadden Sea, albeit at the cost of lower survival especially in years of low lugworm densities.

Even though godwits were able to compensate for the reduced refuelling time by increasing refuelling rates, these rates were insufficient in years when lugworm abundance was low. In such years, godwits left the Wadden Sea with lower body stores that compromised their subsequent survival probability. Females were more sensitive to the shorter refuelling times than males, perhaps because they are larger and have higher energetic requirements such as the need to produce eggs after arrival on the tundra breeding grounds[19]. According to our calculations, refuelling females need lugworm densities 2.2 times higher than the average observed in the Wadden Sea over the last two decades to fully compensate for the observed shorter staging duration (Fig. 2h).

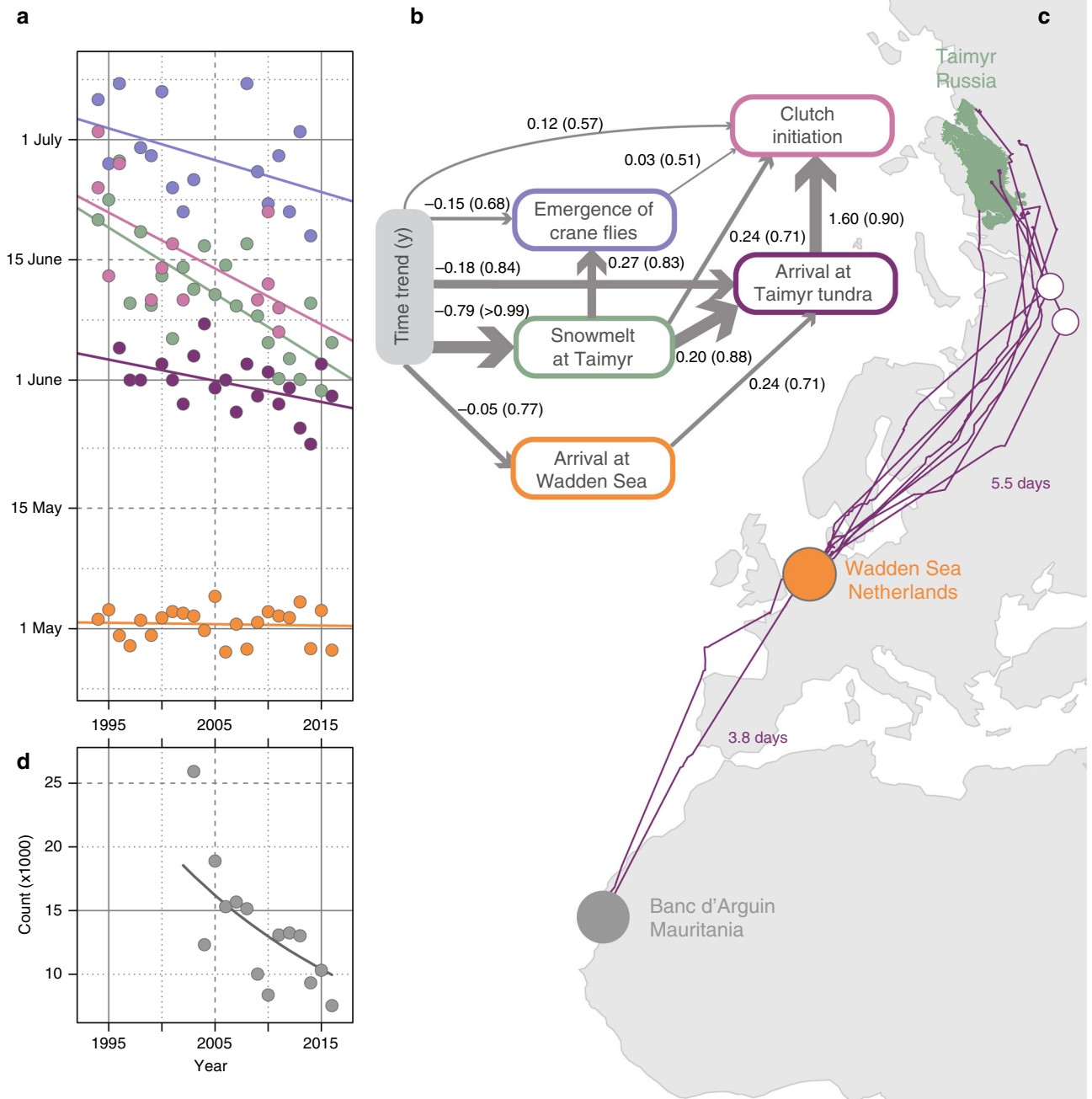

**Fig. 1** The effect of advance in Arctic phenology on spring schedules and, possibly, population dynamics of godwits. **a** Onset of spring (dates of snowmelt, emergence of adult crane fly, arrival of godwits on the tundra breeding area and clutch initiation) at Taimyr Peninsula in the Russian Arctic have advanced, whereas dates of arrival in the Wadden Sea have not. **b** Path analysis revealed that shifts in the dates of the first emergence of crane flies and godwit phenology in the Arctic were mostly driven by changes in the dates of snowmelt. Arrows indicate direction and strength of causal relationships between the variables. Arrows' widths are proportional to the effect strength (coefficient evidence ratios). Estimates of unstandardized path coefficients $\lambda$ and their probabilities $P(|\lambda| > 0)$ (in brackets), from the structural equation model are indicated above the corresponding arrows. Other information on coefficients uncertainty is summarized in Supplementary Table 1. Variable Time represents linear temporal trend. Values for Time are measured over years, while all remaining variables are presented on a daily scale. **c** After the wintering period in West Africa, godwits migrate to the breeding grounds with a single refuelling stop in the Wadden Sea. The stopover lasts on average 24.5 ± 4 days. Violet lines represent spring migratory tracks of eight godwits equipped with satellite transmitters in 2016, together with the estimated duration of migration paths between Banc d'Arguin and Wadden Sea (3.8 d) and between Wadden Sea and Taimyr (5.5 d), and the two white circles show the only additional stopovers (of 2.5 d, each) on approach to the breeding grounds. **d** Counts at one of the major wintering areas of godwits at Banc d'Arguin, Mauritania, West Africa, show a decline in population size. The world borders shapefile used to make this figure was downloaded from Thematic Mapping API (http://thematicmapping.org/downloads/world_borders.php), which is available under a CC-BY-SA license. All rights reserved

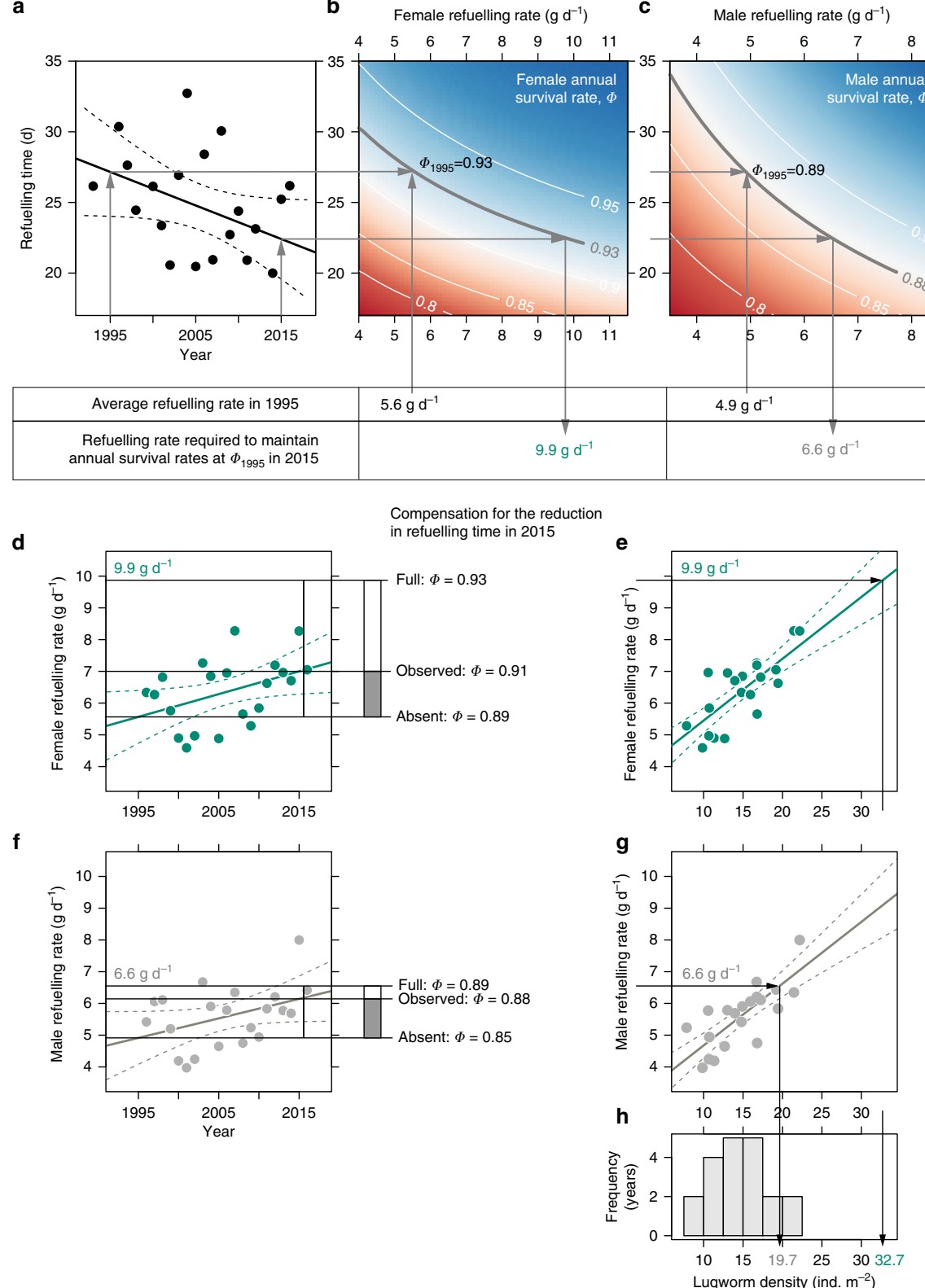

**Fig. 2** Relationships between refuelling time and refuelling conditions and the subsequent annual survival of godwits. **a** The refuelling time in the Wadden Sea shortened between 1995 and 2015. Apparent annual survival of godwits depended on refuelling time and refuelling rate in the Wadden Sea (**b**, for females and **c**, for males) and, therefore, the refuelling rate required to maintain annual survival at the 1995 level has increased substantially (6.6 g d⁻¹ rather than 4.9 g d⁻¹ in males and 9.9 g/d rather than 5.6 g d⁻¹ in females). Godwits partially offset staging time loss by increasing their refuelling rates (**d** for females and **f** for males). Refuelling rates correlated with density of adult lugworms (**e** and **g**). **h** Lugworm densities in the Wadden Sea satisfied increased refuelling demands of male but not female godwits

The refuelling conditions in the Wadden Sea are critical for godwits to cope with earlier snowmelt on the Arctic breeding grounds. Even though refuelling rates are determined not only by food availability, but may be limited physiologically[20,21] and by other environmental factors, such as disturbance rates[22], improvement of food stocks at staging sites can increase survival rates in godwits. Thus, to mitigate negative climate-change effects on godwit population travelling through the Wadden Sea, we need to maintain fuelling conditions for them. As a first step we may suggest the suspension of mechanical lugworm harvesting practices in the Dutch Wadden Sea[23].

The population of bar-tailed godwits we studied is just one of many long-distance migrant birds challenged by the rapid global warming of the Arctic[24]. Food-related limitations on refuelling rate are likely to be a common problem in populations that need to keep up with advancing springs. It is a sobering realization that such refuelling areas are poorly protected in some parts of the world[25] with many being entirely lost or reduced in quality by urban and industrial developments[26–28]. Proactive, international collaborations focused on maximizing resources at staging sites for Arctic breeding migratory birds could help maintain the connectivity between the worlds' variably changing biomes.

## Methods

**Snowmelt dates from remote sensing data.** We used the NOAA Climate Data Record (CDR) estimates of extent of snow cover on a $100 \times 100$ km grid based on remote sensing data[10]. Weekly snow cover for 1992–2017 was overlain with the breeding range of *taymyrensis* bar-tailed godwit[29]. For each grid cell snowmelt date was estimated as the next day after continuous snow cover period. Annual snowmelt date for the breeding range was estimated as the mean date across all grid cells and annual snowmelt date at the breeding site (South-Eastern Taimyr) was estimated as the mean of the two closest NOAA grid cells. Because range-wide trend in snowmelt dates did not significantly differ from the trend at the field site (the slope of the linear regression of overall snowmelt date on that at South-Eastern Taimyr did not differ significantly from unity $1.00 \pm 0.76$, $t = 1.32$, d.f. = 35, $P = 0.20$) we used the snowmelt dates at the site in the analyses below.

NOAA CDR data have estimates of uncertainty within 3–5%, but since snow cover was also recorded daily at the field site[30] (72.8°N, 106.0°E, Supplementary Table 1), we checked whether remote sensing data matched our field observations. The slope of the linear regression for the two NOAA grid cells closest to the field site on the snowmelt dates derived from field measurements did not differ significantly from unity ($0.97 \pm 0.129$, $t = -0.2$, d.f. = 16, $P = 0.84$), and the intercept ($0.18 \pm 1.05$) did not differ significantly from 0 ($t = 0.17$, d.f. = 16, $P = 0.87$). Details on snow data processing and analysis are available as Supporting information[31].

**Data collection on the breeding grounds at Taimyr Peninsula.** First arrival dates of godwits on the breeding grounds were monitored daily near the village of Khatanga, 72.0°N 102.5°E by AG and other staff researchers of Taimyrskiy Nature Reserve between 1992 and 2016. Dates of clutch initiation by godwits and of the first appearance of the adult crane flies (*Tipula* sp.) were recorded by MS, ER and other participants of the Taimyr shorebirds monitoring project at a different location at South-Eastern Taimyr (72.8°N, 106.0°E). A three square km area was systematically searched for nests in each year. Clutch initiation dates were determined either by egg flotation[32] or back-calculation from recorded hatching dates. Clutch initiation dates for all 13 nests found and all phenological data collected at the Taimyr are presented in Supplementary Table 1. Phenological trends over time were determined as slopes of linear regressions of the observed dates over years. All data and analysis code are available as Supporting information[31].

**Structural equations modelling of the phenology data.** To estimate the statistical causality among the observed phenological variables we used path analysis, a special case of structural equations modelling framework[33,34]. In the proposed model we estimated the strengths of the potential causal relationships between phenological variables (dates of arrival to the Wadden Sea, arrival to Taimyr, clutch initiation, snowmelt, and crane fly emergence). The model contained only one independent variable, time (year). The model structure is presented in Fig. 1b. Variables were assumed to have latent state and variable-specific normally

distributed errors:

$$(\text{Arrival Wadden Sea})_i = b_{1.1} \times \text{Time}_i + \varepsilon_i; \varepsilon_i \in \text{Norm}\left(0, \sigma_{\text{arrWS}}^2\right). \quad (1)$$

$$\text{Snowmelt}_i = b_{2.1} \times \text{Time}_i + \varepsilon_i; \varepsilon_i \in \text{Norm}\left(0, \sigma_{\text{snowmelt}}^2\right). \quad (2)$$

$$(\text{Arrival Taimyr})_i = b_{3.1} \times \text{Time}_i + b_{3.2} \times \text{Snowmelt}_i + b_{3.3}$$
$$\times (\text{Arrival Wadden Sea})_i + \varepsilon_i; \varepsilon_i \in \text{Norm}\left(0, \sigma_{\text{arrivalT}}^2\right) \quad (3)$$

$$(\text{Crane fly emergence})_i = b_{4.1} \times \text{Time}_i + b_{4.2}$$
$$\times \text{Snowmelt}_i + \varepsilon_i; \varepsilon_i \in \text{Norm}\left(0, \sigma_{\text{cranefly}}^2\right) \quad (4)$$

$$(\text{Clutch initiation})_i = b_{5.1} \times \text{Time}_i + b_{5.2} \times \text{Snowmelt}_i +$$
$$b_{5.3} \times (\text{Arrival Taimyr})_i + b_{5.4} \times (\text{Crane fly emergence})_i$$
$$+ \varepsilon_i; \varepsilon_i \in \text{Norm}\left(0, \sigma_{\text{clutch}}^2\right) \quad (5)$$

Arrival dates in the Wadden Sea were hypothesised to affect arrival dates to Taimyr, and snowmelt dates on Taimyr to affect all phenological events except for arrival dates in the Wadden Sea. Clutch initiation dates were hypothesised to depend on all variables except arrivals in the Wadden Sea. We used mean estimates for dates of arrivals to Taimyr and snowmelt without uncertainties to ease their combination with other phenological observations. All the dates were centred to have zero mean but not scaled. Effects of variables were estimated as maximum probability of parameter to be strictly positive or strictly negative. The model parameters were estimated with MCMC JAGS sampler[35] via the R2jags[36] interface from the R computing environment[37], the model data and code are available in supporting information.

**Spring migration parameters from satellite tracking data.** In 2016 we deployed 5-g solar-powered Argos PTT-100 transmitters (Microwave Telemetry Inc., Maryland, USA) on eight female godwits (two on the wintering grounds in Mauritania and six during the refuelling period in the Wadden Sea). We used leg-loop harnesses weighing ca. 1 g, similar to the ones used successfully for black-tailed godwits (*Limosa limosa*)[38]. The total attachment mass was ca. 1.2% of the body mass for females departing from the Wadden Sea and 1.5% for females departing from Mauritania[6]. We tracked birds via the Argos system (CLS France, http://www.argos-system.org/) and removed occasional outliers in the Argos data with a hybrid filter[39]. We estimated migration duration as total time that it took a bird to get from wintering to refuelling sites and from refuelling sites to the breeding grounds. Stopovers were defined as time periods during which the birds were located within a 25-km radius for at least 24 h. The northward migratory tracks of the eight godwits are shown in Fig. 1c.

**Arrival dates to the Wadden Sea using citizen science data.** The arrival of godwits to the Dutch coastline has been monitored by a citizen science project http://www.trektellen.nl from 1992 to 2016. In this project, experienced observers count migrating birds at established sites along the Dutch coast[40]. To estimate mean godwit arrival date, we used observations between 10 April and 25 May (the period when godwits are known to arrive[8]) from sites that had over 100 records of at least a single godwit ($n = 7$). The final dataset contained almost 400,000 godwits recorded during 2318 counting sessions.

To estimate annual mean arrival date to the Wadden Sea $\text{T0}_k$, we extended the model by Lindén and Mäntyniemi[41]. We assumed that at year $k$ true daily number of arriving godwits $N_{ijk}$ is distributed normally over days with mean arrival date $\text{T0}_k$ and standard deviation $\sigma \text{T0}_k$. Each observation site $i$ has its own multiplicative effect on the number of godwits that does not shift $\text{T0}_k$. Counts at the sites are proportional to the daily arrival but are outcomes from a random observation process with negative binomial error.

The expected number of birds $N_{ijk}$ on a day $j$ at a site $i$ and a year $k$ can be calculated then with the following equation:

$$\log\left(N_{ijk}\right) = a_0 + \text{total}_k + \text{site}_i + \text{offset}_{ijk}$$
$$- \frac{\left(T_{ijk} - \text{T0}_k\right)^2}{2\sigma \text{TO}_k^2} - 0.5 \log(2\pi) - \log(\sigma \text{TO}_k) \quad (6)$$

where $T_{ijk}$ is the observation day; $\text{offset}_{ijk}$ is the natural log of observation duration (in hours); $a_0$ is the overall baseline – natural log of average number of birds passing through an average site in average year; $\text{total}_k$ is the annual random effect; $\text{site}_i$ is the random effect for observation site.

The observed count $Y_{ijk}$ was assumed to be a result of a random observation process with the error following the negative binomial distribution.

$$Y_{ijk} \in \text{NegBin}\left(p_{ijk}, r\right); p_{ijk} = \frac{r}{r + N_{ijk}}. \qquad (7)$$

Godwits arrive in small flocks[42] and, therefore, the negative binomial distribution of counts was preferred over Poisson or quasi-Poisson distributions[41]. The model was estimated with JAGS sampler[35] via the R2jags[36] interface from the R computing environment[37] and reasonably fitted the data[43] (Bayesian $P$-value 0.59).

**Data collection on food abundance in the Wadden Sea**. We obtained lugworm (*Arenicola marina*) densities from 1996 to 2016 on the basis of the biannual sampling effort at 15 permanent sampling stations located at Balgzand in the western Dutch Wadden Sea[44,45] (52.9°N, 4.8°E). We used the densities of adult lugworms in late winter (Feb–Mar, mean of all stations) as a proxy of their abundance in April and May[46].

During refuelling in the Wadden Sea, most godwits were captured, colour-marked, and resighted near Terschelling island (53.40°N, 5.34°E), 50 km from the lugworm sampling area at Balgzand[8]. To justify the use of the Balgzand data for statistically explaining refuelling rates of the Terschelling-captured godwits, we compared lugworm densities between these two areas using data from 2008 to 2014 collected by the Synoptic Intertidal Benthic Survey (SIBES) program, a large-scale grid sampling of benthic fauna in the Wadden Sea[47,48]. We selected stations within 500 m of the shoreline of Terschelling, as most godwits fuel within this range[8], and compared mean lugworms densities for each year between areas. Lugworm densities were highly correlated (Pearson's $r = 0.86$) between the two areas.

**Data collection on refuelling godwits in the Wadden Sea**. To estimate population-level refuelling rates of godwits in the Wadden Sea, we used sex and body mass records from the birds captured at the two main sites: Castricum and the Dutch Wadden Sea. Godwits arriving from wintering areas in West Africa migrate along the Dutch coast before landing in the Wadden Sea. With song playbacks and decoys, overflying birds were lured into landing at a site near Castricum and captured with modified double clapnets for finches powered by elastic cords. There are no foraging areas for migrating godwits near the Castricum catching site thus birds caught there provided samples representative of migrants arriving after a long non-stop flight[49]. Between 1992 and 2016, members of the Castricum Ringing Group captured, ringed and measured 2722 adult godwits[50]. These data were used to estimate annual sex-specific body mass at arrival. Between 1992 and 2016, 6251 refuelling adult godwits were captured across the Dutch Wadden Sea (4.74–6.21°E), mostly around high tide, with wind driven and pulled wilsternets[9].

**Arrival mass and refuelling rates and time**. The details on how we estimated annual refuelling time and refuelling rate are outlined in Fig. 3. With the measurements obtained from arriving godwits captured at the Castricum ringing station we estimated mean arrival mass of godwits $\text{W0}_{ks}$ for each year $k$ and sex $s$ and sex-specific standard deviation of residuals $\sigma \text{W0}_s$ using lme4 package[51].

Refuelling time ($\text{RT}_k$) was obtained as the difference between year-specific times of departure ($\text{TD}_k$) from and arrival ($\text{T0}_k$) to the Wadden Sea, Fig. 3a. $\text{TD}_k$ was calculated as observed arrival dates at the breeding grounds on the Taimyr Peninsula minus 5.5 days (estimated duration of migration from satellite telemetry data; see Results) and $\text{T0}_k$ is the arrival date to the Wadden Sea, estimated with the arrival model (eq. 6).

Godwits are sexually dimorphic, with males being smaller than females, and thus population-level refuelling rates $\alpha_{ks}$ were modelled for each year $k$ and sex $s$ using the yearly arrival dates ($\text{T0}_k$), arrival mass at Castricum $\text{W0}_{ks}$ (Fig. 3b and d) and the mass of refuelling godwits captured in the Wadden Sea (Fig. 3c and e). We also assessed linear effects of abundance of lugworms on refuelling rates:

$$\mu\alpha_{ks} = \text{Lugworms\_intercept}_s + \text{Lugworms\_slope}_s \times \text{Lugworms density}_k, \qquad (8)$$

$$\alpha_{ks} \in \text{Norm}\left(\mu\alpha_{ks}, \sigma\alpha_s^2\right), \qquad (9)$$

where $\mu\alpha_{ks}$ is the expected and $\alpha_{ks}$ are realized refuelling rates.

Individual mass at capture in the Wadden Sea $W_i$ at Julian day of capture ($T_i$) was assumed to be influenced by arrival mass $\text{W0}_{ks}$, the product of refuelling rate $\alpha_{ks}$, and the time since arrival $T_i - \text{T0}_k$:

$$W_i = \text{W0}_{iks} + \alpha_{ks} \times \left(T_i - \text{T0}_{ik}\right), \qquad (10)$$

where

$$\text{W0}_{iks} \in \text{Norm}\left(\text{W0}_{ks}, \sigma\text{W0}_s^2\right); \text{T0}_{ik} \in \text{Norm}\left(\text{T0}_k, \sigma\text{T0}_k^2\right). \qquad (11)$$

Original data and model predictions are presented in Supplementary Fig. 1 and 2.

**Effects of refuelling time and rate on survival probability**. To estimate effects of departure fuel load on subsequent survival, we individually colour-marked and resighted godwits in the Wadden Sea from 2001[9,52]. We used data from adult birds captured in April–May in The Netherlands in 2002–2016 (3995 individuals). Of these birds, 2252 individuals were resighted at least once, yielding a total of 4813 resightings. It has been suggested that departure fuel load will affect subsequent survival (and even reproduction) in migratory birds[53]. During the spring refuelling period godwits are time limited[7], hence the amount of fuel stored for migration is a product of the refuelling time and refuelling rate[13]. We estimated the effect of fuel load on subsequent survival in the migrants by regressing annual survival $\Phi_{ik}$ over year-specific mean refuelling time ($\text{RT}_k$) and mean refuelling rate ($\mu\alpha_{ks}$, eq. 3). To model multiplicative effect of these two variables in the additive framework we used their natural logarithms.

We used the Cormack–Jolly–Seber (CJS) model to estimate apparent survival probability independently from the resighting probability[54]. Survival probability $\Phi_k$ was modelled as a function of time since marking (TSM, with two classes—'first year after marking' and 'later years'), sex, refuelling rate and refuelling time and their two way interactions. We used mean estimated refuelling rates and durations without accounting for their uncertainty. Because refuelling rate and time are multiplicative we used their natural logarithms to model them in an additive framework:

$$\text{logit}(\Phi_{ik}) \sim \text{TSM}_{ik} + \text{sex}_i + \log(\alpha_{ks}) + \log(\text{RT}_k). \qquad (12)$$

Resighting probability $P$ was modelled as a function of year

$$\text{logit}(P_{ik}) \sim \text{year}_k. \qquad (13)$$

The set of nested models was estimated in program MARK[55], using the RMark[56] interface. The model results were corrected for overdispersion of $1.118 \pm 0.005$, estimated by the median c-hat test in program MARK. Apparent survival probabilities from the CJS model differ from true survival probabilities as they include permanent emigration, but in the case of the godwits, absence of refuelling sites other than the Wadden Sea and thus absence of emigration opportunities makes these two probabilities the same.

**Godwit population trends from winter counts in West Africa**. Every November or December, 2002–2016, we counted godwits at six high-tide roosts near the village of Iwik in Parc National du Banc d'Arguin, Mauritania (19.8°N, 16.3°W). These counts represent only a portion of the wintering population, and exchanges of individuals may occur with other wintering sites along the African coastline[52,57]. The wintering population counts had high variance not permitting integration with annual survival probabilities, so we used them only to model time-independent population growth $\lambda$. Thus, we modelled the change in local numbers in a state-space model similar to the Kéry & Schaub approach[43]. The approach assumes that there were only random, but no systematic, shifts in the distribution over the observation period (for which there is no evidence[57]).

$$N_{i,k+1} = N_{i,k} \times \lambda. \qquad (14)$$

As in equation 2, we assumed a random observation process error following a negative binomial distribution but with a site-dependent error:

$$Y_{ik} \in \text{NegBin}(p_{ik}, r_i); p_{ik} = \frac{r_i}{r_i + N_{i,k}}. \qquad (15)$$

A negative binomial error distribution was chosen to allow for overdispersion, as birds were counted at high tide roosts in large flocks[41].

**Ethical statement**. This research complied with the ethical guidelines of the Dutch law on animal experiments, and was supervised by the Dutch Central Committee for Animal Experimentation (CCD) under guidance of protocol AVD8020020171505 to NIOZ.

**Code availability**. All data used, R code to perform statistical analyses and model results[31] are maintained on GitHub (https://github.com/eldarrak/Godwits_worms_and_climate_change).

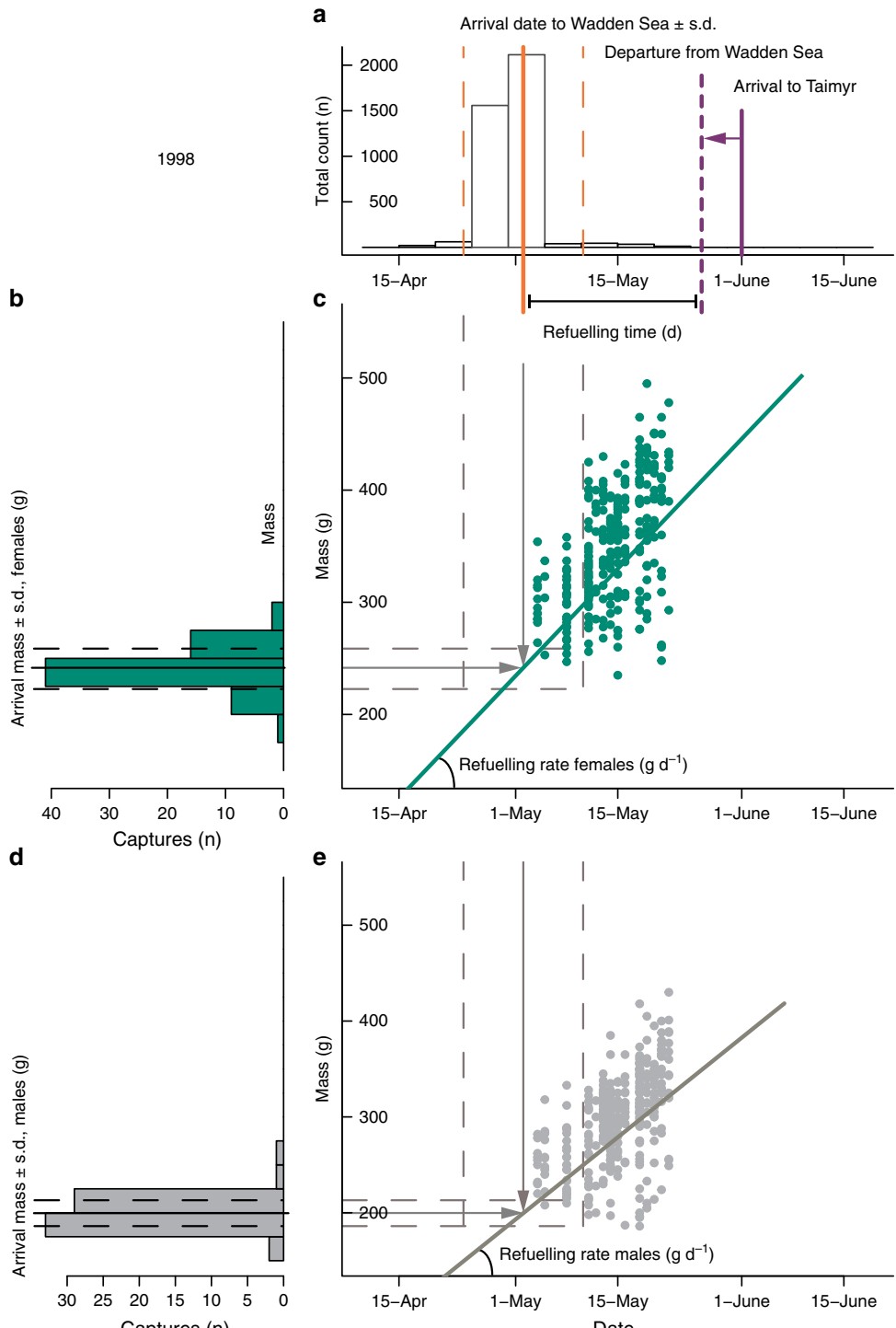

**Fig. 3** Estimation of refuelling time and refuelling rate of godwits in the Wadden Sea for a sample year. **a** Annual refuelling time for year $k$ ($k = 1998$ in the figure) $RT_k$ in the Wadden Sea was estimated as a difference between average arrival and departure dates. Mean arrival date to the Wadden Sea $T0_k$, and its standard deviation $\sigma T0_k$ were estimated from citizen science data on arrival date accounting for observation duration and for variation in observation efficiency between observation sites. Dates of departure from the Wadden Sea were obtained by subtracting the estimated time taken by the migration between Wadden Sea and Taimyr (5.5 days) from dates of first arrival at Taimyr. **b** Annual arrival mass for females $W0_{k\,female}$ was estimated from godwits captured immediately upon arrival from West Africa birds in Castricum. **c** Population-level female annual refuelling rate $\alpha_{k\,female}$ estimation combined arrival date $T0_k$ and arrival mass $W0_{k\,female}$ estimates with body mass values obtained from godwits refuelling in the Wadden Sea. **d** For males, arrival mass $W0_{k\,male}$ and **e** refuelling rate $\alpha_{k\,male}$ were estimated separately as males fuel up slower but are lighter and need to accumulate less fuel for migration

## Data availability

All data and model results[31] are maintained on GitHub (https://github.com/eldarrak/Godwits_worms_and_climate_change).

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

## Acknowledgements

We thank Viktor Golovnyuk in Russia, and Maarten Brugge and Bernard Spaans in The Netherlands and in Mauritania, for their roles in the field work on godwits; Cathrinus Monkel and other wilsterflappers for help with bird catching; Harry Horn, Jan de Jong and Jacob de Vries and others for their efforts to resight colour-marked godwits; Jan Beukema for establishing the long-term sampling of lugworm densities; all observers submitting observations to trektellen.nl database and Gerard Troost for curating this database and providing the data; P.W.N. for providing access to VRS Castricum to the dune area. We thank Bart Kempenaers, Res Altwegg and Diego Rubolini for constructive comments on the manuscript. E.R. and T.P. were supported by grants from the Waddenfonds (Metawad, WF209925) and the Common Wadden Sea Secretariat, Wilhelmshaven, Germany. Field work in Russia was supported by National Park Schleswig-Holstein and State Nature Reserve Taimyrskiy, and the field work in the Netherlands and Mauritania by NIOZ and the Waddenfonds Metawad project (WF209925). The satellite tracking work, and the contribution of J.K., were financed by the Spinoza Premium 2014 to T.P. of the Netherlands Organisation for Scientific Research (NWO). Any use of trade, firm, or product names is for descriptive purposes only and does not imply endorsement by the U.S. Government.

## Author contributions

T.P. conceived the study and made sure the field work in the Wadden Sea and at Banc d'Arguin kept on going. E.R., S.D., C.J.C., VRS Castricum, A.D., R.D., A.G., J.t.H., J.J., M.S., T.L.T., J.A.v.G. and T.P. collected primary data. S.D. and J.t.H. curated the godwit capture data, for a long time C.J.C. curated citizen science bird count data. After initial steps by S.D., E.R. analysed the data with help of A.S. E.R., S.D., J.A.v.G. and T.P. outlined the paper. E.R. and T.P. wrote the paper, which was improved by J.A.v.G., S.D. and J.K. All authors discussed the results and commented on the manuscript.

## Additional information

**Competing interests:** The authors declare no competing interests.

## VRS Castricum

Andre van Loon[9], Arnold Wijker[9], Guido Keijl[9], Henk Levering[9],  Jan Visser[9], Leo Heemskerk[9], Luc Knijnsberg[9], Marc van Roomen[9], Paul Ruiters[9], Piet Admiraal[9], Piet Veldt[9], Richard Reijnders[9] & Walter Beentjes[9]

[9]VRS Castricum—Bergstraat 39, 1931 EN Egmond aan Zee, The Netherlands

