## [Peer Review File · Nature Communications]

Reviewers' comments:

Reviewer #1 (Remarks to the Author):

I liked reading this manuscript. You managed to combine a lot of different data sets that together tell a compelling story. I appreciate the fact that you made all your code available. My main concern is that the statistical methods section should be more detailed. Readers should be able to understand what was done without having to work through code.

Line 210: from your description, it is not entirely clear what you are trying to estimate: I'm not sure what you mean by "true daily arrival date". Perhaps "true number of individuals arriving daily"? But you seem to model the number of encountered birds N_{ijk} . From my reading of your equation 1, it looks like you are assuming that the number of godwits over time follows a bell-shaped curve: they increase as more individuals arrive and then decrease again as individuals leave. The main parameter you seem to be interested in is $T0_k$, which I understand is the time at which godwit density (number of birds encountered per hour of observation) is highest. Is this what you interpret as 'time of spring arrival'? Please clarify this section.

Description of the structural equation model needs a bit more detail. What is 'time' in Supplementary Figure 2? Is it a temporal trend? Is clutch initiation what you call breeding in the code? Please describe the variables and model structure more carefully. In my understanding, 'causality' in SEMs comes from the causal assumptions you make, i.e. the model structure. So it is particularly important that you describe the model and its inputs well. And I feel you need to be a bit more careful in the manuscript how you interpret your results as causal relationships. Another thing that was unclear to me is how you accounted for the uncertainty in the variables in this model. E.g. $T0_k$ is a key variable. Since these are estimates rather than observations, how did you account for the uncertainty? At least, this needs to be mentioned.

Data collection on refuelling rates (lines 232 and following): again here I'm missing some key detail. You describe that godwits were caught and that you estimated 'body mass at arrival'. How did you know when these birds arrived? If you caught unmarked birds, they might have been at the site for a while before you caught them. And how did you get from body mass to refuelling rates? Did you capture the same individuals multiple times? Or is this based on a population-level relationship between body mass and time of the season? Please clarify.

Refuelling rates (lines 243 and following): this also needs more detail. I understand that α_{ks} is the refuelling rate. What is μ ? What is K_{sat_s} in equation 3? 'Arrival mass' depends on the assumption that the birds captured by the Catricum group were indeed freshly arriving birds. How good is this assumption and how sensitive are your results to this assumption? In equation 6, I assume the μ and σ for $T0$ comes from the earlier analysis of arrival date. How did you estimate the μ and σ for $W0$?

Equation 7: should be ' $\text{logit}(\Psi_{ik}) = \dots$ '

Equation 8: same here, you are modelling $\text{logit}(P)$, I presume.

The survival analysis did not account for the uncertainty in the covariates (which were estimated rather than observed). That's ok but should be acknowledged.

Discussion: I feel it is a bit a strong assertion to say that the godwits did not depart from their wintering grounds earlier since you have no direct data on that. All you have is no evidence that timing at their refuelling site changed. Similarly, I feel that your assertion that they shortened the

staging period is too strong. This depends on the observed earlier arrival and also the assumption of constant travelling times between the staging and breeding areas.

Res Altwegg

Reviewer #2 (Remarks to the Author):

The ms by Rakhimberdiev et alii reports on climate change effects on the migratory behavior of a long-distance migratory bird, the bar-tailed godwit. By merging long-term data from different continents, the authors convincingly show that this polar wader species is advancing its breeding and arrival phenology in response to advancing phenology in the Arctic, and that this can be achieved by shortening staging time in their only spring migratory fuelling area in the Netherlands. However, this comes at a fitness cost, because the shorter the fuelling time, the lower the survival of godwits. There is no evidence instead that the species is showing advanced departure from African wintering grounds, as gauged from lack of advanced arrival to the Netherlands. Hence, by integrating observations from fuelling individuals, this important study suggests that maintaining high-quality fuelling grounds in the Netherlands is crucial for long-term conservation efforts as it can possibly buffer negative climate change effects on this species. However, there is evidence for long-term population decline, which may suggest that current conservation status of fuelling grounds is not enough to buffer negative climate change effects on this species.

I will report my comments in order of appearance, and mark major issues with M.

L25: we show that when

L26: variation

L30: This implies

L37: events, such as onset of spring, that has been advancing in recent decades in most temperate and boreal ecosystems.

L37-40: wording needs to be improved here

L41: changes, resulting in low reproductive output, and thus population decline.

L43: about the fitness costs that such adjustments imply. We expect such costs

L45: annual cycles use widely different regions of the globe with often drastically

L46: timing with advancing

L48: Godwits are among the several long-distance migrant wader species that travel ...with a single refuelling stop (of ca. XX days) in...

L51: and use most of

L68: godwits over XX years. (note that you have mentioned years for all other measures! So mention here and not in the first line of the Results)

L71: The phenology of ...

L71: here and elsewhere: please specify whether an association is statistically significant or not. E.g. ...on Taimyr has significantly advanced

L71: here and elsewhere. Reporting 'naked' p-values is considered a poor statistical practice. You must report, besides the estimate and SE, also the test statistic (e.g. t, F) and associated d.f..

L72: here and elsewhere: I suggest to report annual changes in days/year, as in most phenological studies dealing with birds. This is indeed a common metric of those studies, and would make comparisons easier.

L72: please state here that all error measures are SE (better s.e.m. in Nature style) throughout, otherwise always add s.e.m. along all error estimates. Also, symbols should be spaced out ($P = 0.001$ instead of $P=0.001$).

L73-74: this is unclear until one reads the Methods. I suggest removing the last part referring to the field study site (estimate is the same as the whole breeding range). I will comment more on this in the Methods section.

L74: Crane flies responded....: this is very unclear until the Methods and more details are needed. What crane flies population parameters were examined? Briefly mention that you analyse first emergence time, here and in the Legend to Fig. 1.

L78: briefly mention which kind of analysis you refer to here. Structural equation modelling? Path analysis? I think supplementary figure 2 should be reported in the main manuscript body.

L82: previously suggested

L86: did not significantly

M L86-89: this part of the analyses sounds very unclear to me. I find it very odd that you regress fuelling time (occurring at time i) on snowmelt date (occurring at time $i+x$). Does it make any biological sense? How can you predict something that occurred in the past from an event occurring in the future? What is the expected direct causal association between the two variables? This should be clearly explained already here. Actually, based on what you write (As the time of arrival...did not change), I would have expected a (lack of) significant temporal trend in arrival date to the NL here. To me it would seem more appropriate to show temporal trends in arrival date to the NL here and in time available for refuelling. These should be reported. Also, the variable 'time available for refuelling' is vague and unclear what exactly it is. A brief description is needed here.

From a causal point of view, I understand the following flow: time of arrival to the Netherlands has not changed through time. However, time of breeding has advanced through time. The latter is achieved by significantly shortening the refueling stopover duration in the NL through time. The predictor is always 'year', not date of snowmelt (which by the way has advanced through time).

Please clarify or change.

M L91: all the subsequent analyses are obscure and poorly presented. Up to this point, you have reported significance values, estimates and slopes for each statistical analysis (though test statistics were missing). From here onward, a lot of statistical analyses are made, but it is not clear at all where

do they come from or how they were realized. Nor it is clear exactly which analyses have been performed (see other comments below, more in detail). I really think that the manuscript would benefit from a summary table reporting all the temporal trends and analysis made, including parameters estimates, test statistics, and P-values. By this way, you would be able to remove most statistics from the main text, improving readability.

M L92-93: here for the first time, males and females are analyzed separately. This is confusing, as the scale of this analysis is in my opinion too broad to differentiate between the sexes. Up to here sex differences are not mentioned. The reasons for differentiating males and females should be briefly but effectively laid out here.

L94-95: such sex differences are surprising: are these real? This means that males almost completely compensated for, while this was not the case for females. Is the sex difference significant? I believe not, but where does these hugely different percentage compensations originate from? Also, replace 'partially compensated' with 'partly offset' or the like.

L96: the unit measurement of these slopes is confusing: are these temporal trends in refuelling rates? If so, I would rather express these as (g/d)/year. I know it is the same, but as this it looks more straightforward.

L99-100: the last two sentences are really puzzling: decrease in survival over time? Significant? How strong (no slopes provided) is it? How exactly did it contribute to the yearly population decline in Mauritania (by the way: how derived? Significant?)? Did you perform any formal analysis? Is it just a consequence you have derived a conclusion from two separate results (i.e. the temporal decline in survival and the negative temporal population trend)? In general, I strongly advise carefully revising the whole text of the result from L91-100.

L104: birds deal with

L105-107: very poorly structured, rephrase and be more accurate. 'Victim' is colloquial jargon, should be avoided. 'Trophic mismatch' should be clearly spelled out as desynchronization between peak food demand and peak seasonal food availability. 'Re-synchronize' is unclear to me: I would rather say something like: 'godwits appear able to keep track of the temporal advancement of their main...'

L108-109: of course I understand what do you mean. But be careful because you did NOT assess departure from the African wintering grounds! Consider rephrasing this sentence.

L112: Although rigid... (delete comma).

L112-115: This sentence is poorly worded and unclear, rephrase.

L114 and L116: 'In the present case' repeated twice in consecutive sentences, modify text.

L123: 'squeezed' is poorly worded and colloquial jargon. Avoid and rephrase.

L134-136: rephrase sentence to make reasoning more clear. 'Is finite' looks unclear to me in this context. Here, you may also cite the recent article by Studds et al Nat Commun on population collapse in EAAF waders, where it is suggested that environmental changes at the stopover areas in the Yellow Sea are negatively impacting EAAF wader populations.

L149: In general the subheadings of the methods are not very well conceived, not well organized, and

do not describe the content very well. They should be general but informative, and allow quick reference to the Results. The order should closely match that of the results. Please look at these carefully.

L152: state the year range

L155: please report here, rather than in the results section, the details of the analyses carried out on the two NOAA grid cells close to the study site, and say that the estimate for these two cells is very similar to the estimate obtained for the entire range.

L164: please add here the mean percentage of body mass of the devices (including harness) for the tracked birds, and provide details on methods of device attachment to the birds.

L181-183: very unclear. What do you mean? Do you have also 'median' dates? Medians of what distribution? Please clarify or rephrase.

L179: clutch initiation dates: for how many nests/year (give mean and min-max) has this variable been recorded? What was the protocol used? Standardized nest searches in a fixed area? Casual nest findings? Please clarify. How was this variable expressed? Median/mean first egg laying date in each year? Also, instead of 'clutch initiation date', use the more standard term 'first egg laying date'.

L190: late-winter densities: how expressed? Mean values for all stations? This is not specified.

L196: in the two areas: which ones? Clarify. Did you mean between the two areas?

L198-199: unclear how the lugworm densities in Balgzand and in Terschelling were expressed and compared. Mean of all sampling stations for each year at each site?

L209-221: this part could be moved to the supplementary information as it is definitely too much technical for the normal reader. Simply explain the general purpose of the analyses and modelling here.

M L232: paragraph should be merged with the one at L243. In general, this part is very unclear. I am really getting lost here. You must clearly explain the rationale for computing refuelling rates from captures: which data were collected and for which purpose in a very schematic manner. Perhaps a flow chart might be of help. It is really difficult to figure out which kind of analyses have been made and for what purpose. If I understood correctly, you have calculated body mass at arrival for birds sampled with decoys, and refuelling body mass from birds captured at high tide roosts. From this you have computed annual refuelling rate, accounting for lugworm abundance. For each year, arrival date was estimated from citizen science data, and departure date was estimated as first arrival in Taimyr minus the mean number of days taken by satellite-tagged godwits to migrate from NL to Taimyr. The difference between departure and arrival for each year was the stopover duration. But (if I am not wrong) I could understand this only after looking at one of the supplementary figures 2/3! And all these steps are not clearly explained in the manuscript! I think that a representative example (one year) taken from this figures should be reported in the main text to illustrate the method used to derive each variable from which data.

L232: the first sentence does not make sense, strictly speaking: capturing birds is not enough for estimating refuelling rates. Please rephrase. Also, the first trapping method is not explained: were mist-nets used? Only decoys and playback (to attract) are mentioned, but not the capture method.

L239: add comma after 2016.

L244: this is not clear. T0k was not estimated for each sex. If there is sex-specific migration, using a single value for both sexes may lead to erroneous conclusions. The reason for modeling refueling rates separately for each sex should be detailed.

L253: these values appear rather similar: again, is it really necessary to use sex-specific values?

L257: all the description of how refuelling time was calculated should go in the previous paragraph, not here! This paragraph should only deal with the MARK analyses.

L271: 5.5 days: what is the variance around this figure? If it is large, it may not be appropriate to use this value for all data. On how many individuals was it computed?

L282: but in the case

L290: break sentence before 'thus'

Fig.1: add mean duration of the two segments of migration on the graphs (along tracks). Why 95% confidence bands are not shown for panel b? Either remove from all panels or add to all panels.

L509: spring migratory tracks

L516: Legend to figure 2 is confusing and should be carefully revised to improve clarity and readability. Please make a further effort to improve it.

L521-522: refuelling time in the y-axis label, but refuelling duration in the legend. Please be consistent and standardize terminology throughout!

L527: it is not only male survival which is in the supplementary, it is all the male-based analyses. Please clarify this in the legend.

In the beginning of the reply we like to thank the reviewers and also summarise what we have done.

In short, we appreciate:

1. Many suggestions on improvements of the methods section, including:
 - Addition of the Methods section statistical causality analysis;
 - Change of order in the methods section to make it corresponding to the order of Results section;
 - Addition of the Figure 3 into the Methods section, to clarify our modelling approach.
2. Results section was greatly improved by the suggested table with all the details on the results of the analysis. The second half of the Results section was completely rewritten in a cleaner and more logical way.
3. Figure 2 and Figure 3 were redone according to suggestions:
 - Figure 1 is over changed to trends over time, instead of trends over snowmelt;
 - Path analysis results moved to Figure 1 from supplementary materials;
 - Figure 2 was redrawn to ease readability.
4. All the many small textual suggested on the language and formulas were incorporated in the text.

Reviewers' comments:

Reviewer #1 (Remarks to the Author):

I liked reading this manuscript. You managed to combine a lot of different data sets that together tell a compelling story. I appreciate the fact that you made all your code available.

Response: Thank you for taking the time to review our manuscript. We appreciate the feedback you provided. We have addressed your comments below.

My main concern is that the statistical methods section should be more detailed. Readers should be able to understand what was done without having to work through code.

Response: Many thanks! Also for the suggestion. We agree that the methodological part needed attention, and we have done our best to restructure it and make it clearer. We have also added a Figure in the methods section to help people really understand the modelling approaches.

Line 210: from your description, it is not entirely clear what you are trying to estimate: I'm not sure what you mean by "true daily arrival date". Perhaps "true number of individuals arriving daily"? But you seem to model the number of encountered birds N_{ijk} . From my reading of your equation 1, it looks like you are assuming that the number of godwits over time follows a bell-shaped curve: they increase as more individuals arrive and then decrease again as individuals leave. The main parameter you seem to be interested in is $T0_k$, which I understand is the time at which godwit density (number of birds encountered per hour of observation) is highest. Is this what you interpret as 'time of spring arrival'? Please clarify this section.

Response: Yes, you have interpreted it correctly. We think we have improved the model description in the revised manuscript.

Description of the structural equation model needs a bit more detail.

Response: Indeed, details on the path analysis were missing in the manuscript. We have fixed this in the present revision.

What is 'time' in Supplementary Figure 2? Is it a temporal trend?

Response: Yes, time is the temporal trend. Hopefully we have clarified this in the Methods section and in caption of the revised figure. Note that in the revised manuscript this figure became Fig 1b.

Is clutch initiation what you call breeding in the code?

Response: Yes! This is now specified in the supplementary code.

Please describe the variables and model structure more carefully. In my understanding, 'causality' in SEMs comes from the causal assumptions you make, i.e. the model structure. So it is particularly important that you describe the model and its inputs well.

Response: Totally agree. We provide an extended model description in the revised Methods section.

And I feel you need to be a bit more careful in the manuscript how you interpret your results as causal relationships.

Response: Thank you! We have double-checked for the possible overstatements in the Results and made sure to be careful in presenting the results.

Another thing that was unclear to me is how you accounted for the uncertainty in the variables in this model. E.g. $T0_k$ is a key variable. Since these are estimates rather than observations, how did you account for the uncertainty? At least, this needs to be mentioned.

Response: You are right, we did not estimate uncertainty of $T0_k$ separately, it is combined with the shape of the bell-shaped arrival distribution within the parameter $\sigma T0_k$. To clarify our modelling approach we have added Figure 3.

Data collection on refuelling rates (lines 232 and following): again here I'm missing some key detail. You describe that godwits were caught and that you estimated 'body mass at arrival'. How did you know when these birds arrived? If you caught unmarked birds, they might have been at the site for a while before you caught them.

Response: The Castricum catching site is located on the approach of godwits migrating from West-Africa to the Wadden Sea, located after at least 500 km without suitable refuelling stops. Arriving godwits normally do not land there. Using sounds, the birds captured there were lured by catchers, and, can thus be considered as a good representation of freshly arrived godwits. We expanded the explanation in the Methods section and added references.

And how did you get from body mass to refuelling rates? Did you capture the same individuals multiple times? Or is this based on a population-level relationship between body mass and time of the season? Please clarify.

Response: We did not recapture birds. The reported rates are the population-level changes in the body mass over time. We clarify it now in the appropriate sections.

Refuelling rates (lines 243 and following): this also needs more detail. I understand that α_{ks} is the refuelling rate. What is μ ? What is K_{sat_s} in equation 3?

Response: Thanks for noticing! We realized that an explanation of the functional response part of the model was not really necessary given that the refuelling rates observed were far below

the maximum rates possible, and still on the linear increase part of the response curve. In order to streamline the manuscript, we have replaced the functional response model with that of a linear relationship between refuelling rates and lugworm densities and changed the Methods section accordingly.

'Arrival mass' depends on the assumption that the birds captured by the Castricum group were indeed freshly arriving birds. How good is this assumption and how sensitive are your results to this assumption?

Response: We consider this assumption well-supported as en route to this Castricum site for at least 500 km there are no feeding areas used by migrating godwits. Birds caught there provide samples representative of migrants arriving after a long non-stop flight. We improved the appropriate section of the Methods and provide relevant references in support.

In equation 6, I assume the μ and σ for T_0 comes from the earlier analysis of arrival date. How did you estimate the μ and σ for W_0 ?

We have estimated the body mass values upon arrival for each year and the individual variation around these mass values using a simple mixed model. Details on this model were missing in the previous Methods section and were added now. Thanks for noticing.

Equation 7: should be 'logit(Ψ_{ik}) = ...'

Equation 8: same here, you are modelling logit(P), I presume.

Response: Exactly! Corrected, thanks!

The survival analysis did not account for the uncertainty in the covariates (which were estimated rather than observed). That's ok but should be acknowledged.

Response: Good point! We acknowledge this fact in the revised manuscript.

Discussion: I feel it is a bit a strong assertion to say that the godwits did not depart from their wintering grounds earlier since you have no direct data on that. All you have is no evidence that timing at their refuelling site changed. Similarly, I feel that your assertion that they shortened the staging period is too strong. This depends on the observed earlier arrival and also the assumption of constant travelling times between the staging and breeding areas.

Res Altwegg

Response: Thank you for raising the question. You made us think again, whether we can say this. We understand that the statement might seem precarious. Here we provide our reasoning. The migration of bar-tailed godwits in this flyway, from West Africa across Europe to the Siberian breeding grounds, has now been studied since the mid-1980s and the basic inference on the structure of their northward migration based on (1) intense colour ringing and resighting efforts, (2) the analyses of count sequences, and (3) migration observations in all countries along the flyway including Fennoscandia, resulted in the hypothesis that the godwits would make two ca 5000 km long nonstop flights with the only staging period in the Wadden Sea. This makes sense also, as (4) there are few sites *en route* which are suitable for bar-tailed godwits, and this is true both for the Mauritania-Wadden Sea and for the Wadden Sea-Siberia segments of the route; and (5) the accumulated fuel load on departure from the wintering grounds and from the Wadden Sea allows these non-stop migrations. For the nonstop flights variations in travel speeds (caused by e.g. changing wind patterns) are not that important as flight durations are short in comparison with fuelling times (see Hedenström papers etc.).

The evidence above is established and discussed in the few publications we refer to in the manuscript, which makes it unnecessary to bring it up in the Discussion section. Given

that the new satellite tracking data of individual migrations presented in this MS are entirely consistent with this earlier developed view of the migrations, we believe that the inference is robust and we can say that departure timing from West-Africa has remained unchanged while departure timing from the Wadden Sea did change. On this basis we feel that we can indeed make the assertions on constant arrival timing and shortening of refuelling time. Note, however, that we have extended the discussion of the inferences quite a bit and presented it more cautiously in the revised manuscript.

Reviewer #2 (Remarks to the Author):

The ms by Rakhimberdiev et alii reports on climate change effects on the migratory behavior of a long-distance migratory bird, the bar-tailed godwit. By merging long-term data from different continents, the authors convincingly show that this polar wader species is advancing its breeding and arrival phenology in response to advancing phenology in the Arctic, and that this can be achieved by shortening staging time in their only spring migratory fuelling area in the Netherlands. However, this comes at a fitness cost, because the shorter the fuelling time, the lower the survival of godwits. There is no evidence instead that the species is showing advanced departure from African wintering grounds, as gauged from lack of advanced arrival to the Netherlands. Hence, by integrating observations from fuelling individuals, this important study suggests that maintaining high-quality fuelling grounds in the Netherlands is crucial for long-term conservation efforts as it can possibly buffer negative climate change effects on this species. However, there is evidence for long-term population decline, which may suggest that current conservation status of fuelling grounds is not enough to buffer negative climate change effects on this species. I will report my comments in order of appearance, and mark major issues with M.

Response: Thank you for taking the time to review our manuscript. We appreciate the feedback you provided. We have addressed your comments below

L25: we show that when

Response: Corrected, thank you!

L26: variation

Response: Done: "variation in food abundance in the spring staging area"

L30: This implies

Response: Done: "This implies that conditions in the temperate zone determine the ability of godwits to cope with climate-related changes in the Arctic".

L37: events, such as onset of spring, that has been advancing in recent decades in most temperate and boreal ecosystems.

Response: The sentence has been changed, so this edit is not applicable anymore.

L37-40: wording needs to be improved here

Response: Rephrased to "In the Arctic, climatic changes are the strongest, with the highest rates of advance in spring phenology, which is contrasting with the slow changes in equatorial regions."

L41: changes, resulting in low reproductive output, and thus population decline.

Response: we have rephrased this sentence.

L43: about the fitness costs that such adjustments imply. We expect such costs

Response: Good suggestion, thank you!

L45: annual cycles use widely different regions of the globe with often drastically

Response: Done.

L46: timing with advancing

Response: This sentence is no longer part of the revised manuscript

L48: Godwits are among the several long-distance migrant wader species that travel ...with a single refuelling stop (of ca. XX days) in...

Response: Our *taymyrensis* godwits are a subspecies, but not a species, with the other subspecies having very different ranges. We, therefore, rephrased as follows: This population is among several long-distance migratory shorebirds that travel from wintering grounds in West Africa to breed in the Russian Arctic with a single refuelling stop (of ca. 25 days) in ...

L51: and use most of

Response: Done.

L68: godwits over XX years. (note that you have mentioned years for all other measures! So mention here and not in the first line of the Results)

Response: Corrected.

L71: The phenology of ...

Response: this sentence was completely changed, and the suggestion is not applicable anymore.

L71: here and elsewhere: please specify whether an association is statistically significant or not. E.g. ...on Taimyr has significantly advanced

Response: Thank you for the suggestion. We have done it in some places. In some other cases, to simplify already long sentence in cases when we report P values immediately after the association statement, we omitted the 'statistically significant'.

L71: here and elsewhere. Reporting 'naked' p-values is considered a poor statistical practice. You must report, besides the estimate and SE, also the test statistic (e.g. t, F) and associated d.f..

Response: In the revised version we follow your suggestion to M L91 to summarize all the tests in a table and provide all the necessary d.f., n, and test values there.

L72: here and elsewhere: I suggest to report annual changes in days/year, as in most phenological studies dealing with birds. This is indeed a common metric of those studies, and would make comparisons easier.

Response: Very useful suggestion. Many thanks!

L72: please state here that all error measures are SE (better s.e.m. in Nature style) throughout, otherwise always add s.e.m. along all error estimates. Also, symbols should be spaced out ($P = 0.001$ instead of $P=0.001$).

Response: Done, thank you!

L73-74: this is unclear until one reads the Methods. I suggest removing the last part referring to the field study site (estimate is the same as the whole breeding range). I will comment more on this in the Methods section.

Response: Good idea. We have simplified logic here and only use one estimate of date of snowmelt in the Results section together with a short reasoning in the Methods section.

L74: Crane flies responded....: this is very unclear until the Methods and more details are needed. What crane flies population parameters were examined? Briefly mention that you analyse first emergence time, here and in the Legend to Fig. 1.

Response: Done

L78: briefly mention which kind of analysis you refer to here. Structural equation modelling? Path analysis?

Response: Done. We now explain that we use path analysis to infer statistical causality in the phenological data.

I think supplementary figure 2 should be reported in the main manuscript body.

Response: Great idea! Supplementary figure 2 became Figure 1 b in the revised version.

L82: previously suggested

Response: Done. Thank you!

L86: did not significantly

Response: Done. Thanks!

M L86-89: this part of the analyses sounds very unclear to me. I find it very odd that you regress fuelling time (occurring at time i) on snowmelt date (occurring at time $i+x$). Does it make any biological sense? How can you predict something that occurred in the past from an event occurring in the future? What is the expected direct causal association between the two variables? This should be clearly explained already here. Actually, based on what you write (As the time of arrival...did not change), I would have expected a (lack of) significant temporal trend in arrival date to the NL here. To me it would seem more appropriate to show temporal trends in arrival date to the NL here and in time available for refuelling. These should be reported. Also, the variable 'time available for refuelling' is vague and unclear what exactly it is. A brief description is needed here.

From a causal point of view, I understand the following flow: time of arrival to the Netherlands has not changed through time. However, time of breeding has advanced through time. The latter is achieved by significantly shortening the refueling stopover duration in the NL through time. The predictor is always 'year', not date of snowmelt (which by the way has advanced through time).

Please clarify or change.

Response: This makes a lot of sense, thank you! We rewrote this paragraph and changed the corresponding graph as you suggested – we now presents trends over time instead of trends between distant phenological events and dates of snowmelt. All trends over time are also reported now in the new Table 1.

M L91: all the subsequent analyses are obscure and poorly presented. Up to this point, you have reported significance values, estimates and slopes for each statistical analysis (though test statistics were missing). From here onward, a lot of statistical analyses are made, but it is not clear at all where do they come from or how they were realized. Nor it is clear exactly which analyses have been performed (see other comments below, more in detail). I really think that the manuscript would benefit from a summary table reporting all the temporal trends and analysis made, including parameters estimates, test statistics, and P-values. By this way, you would be able to remove most statistics from the main text, improving readability.

Response: Thank you for the advice! We also think that a table summarizing details on the analyses was a great improvement. Please have a look at the Table 1 in the revised manuscript.

M L92-93: here for the first time, males and females are analyzed separately. This is confusing, as the scale of this analysis is in my opinion too broad to differentiate between the sexes. Up to here sex differences are not mentioned. The reasons for differentiating males and females should be briefly but effectively laid out here.

Response: We reworked the explanations for all sex-specific aspects of analyses, added clarifications in the Methods and a figure (Figure 3) to explain how we measured sex-specific responses. We also added discussion on the sex differences in the response to shortened refuelling times.

L94-95: such sex differences are surprising: are these real? This means that males almost completely compensated for, while this was not the case for females. Is the sex difference significant? I believe not, but where does these hugely different percentage compensations originate from? Also, replace 'partially compensated' with 'partly offset' or the like.

Response: The difference between sexes is caused by a strongly sex-specific effect of refuelling time on subsequent survival. We added discussion on the potential causes of the observed effects. We suggest that, because females need to accumulate resources for egg laying, they might be more affected by the shortening of refuelling time. Female godwits are also 17% heavier, so need more fuel to power the nonstop flight to the breeding grounds.

L96: the unit measurement of these slopes is confusing: are these temporal trends in refuelling rates? If so, I would rather express these as (g/d)/year. I know it is the same, but as this it looks more straightforward.

Response: Good idea, but according to the style of the journal we had to use $g\ d^{-1}\ y^{-1}$

L99-100: the last two sentences are really puzzling: decrease in survival over time? Significant? How strong (no slopes provided) is it? How exactly did it contribute to the yearly population decline in Mauritania (by the way: how derived? Significant?)? Did you perform any formal analysis? Is it just a consequence you have derived a conclusion from two separate results (i.e. the temporal decline in survival and the negative temporal population trend)? In general, I strongly advise carefully revising the whole text of the result from L91-100.

Response: We have completely rewritten this part of the Results section, added all the comparisons, slope values and standard errors suggested above, together with a separate table for all test details (Table 1). We did not directly model the changes in population size as a function of changes in survival, as the precision of the population abundance estimates is rather low, and we did not have annual estimates of reproductive success, all necessary for refinement in an integrated model. These points are now made in the appropriate part of the Methods section.

L104: birds deal with

Response: we replaced “birds may deal with” with “birds cope with”

L105-107: very poorly structured, rephrase and be more accurate. ‘Victim’ is colloquial jargon, should be avoided. ‘Trophic mismatch’ should be clearly spelled out as desynchronization between peak food demand and peak seasonal food availability. ‘Re-synchronize’ is unclear to me: I would rather say something like: ‘godwits appear able to keep track of the temporal advancement of their main...’.

Response: we removed jargon and rephrased:

“Rather than the use of multiple sites simply being a liability, it also provides opportunities for among-season compensation. Apparently, in contrast to many other species, bar-tailed godwits adjusted their arrival on the breeding grounds and the onset of breeding, thereby tracking the seasonal advancement of their main arthropod prey on the breeding grounds.”

L108-109: of course I understand what do you mean. But be careful because you did NOT assess departure from the African wintering grounds! Consider rephrasing this sentence.

Response: We have rephrased the sentence and talk only about arrival to the Wadden Sea, but not about departure from West Africa in the revised manuscript.

L112: Although rigid... (delete comma).

Response: Done

L112-115: This sentence is poorly worded and unclear, rephrase.

Response: We have completely rephrased paragraph to improve the clarity of the discussion.

L114 and L116: ‘In the present case’ repeated twice in consecutive sentences, modify text.

Response: Corrected

L123: ‘squeezed’ is poorly worded and colloquial jargon. Avoid and rephrase.

Response: We have streamlined the Discussion, and this unnecessary sentence is no longer there.

L134-136: rephrase sentence to make reasoning more clear. ‘Is finite’ looks unclear to me in this context.

Response: We removed the sentence as unnecessary.

Here, you may also cite the recent article by Studds et al Nat Commun on population collapse in EAAF waders, where it is suggested that environmental changes at the stopover areas in the Yellow Sea are negatively impacting EAAF wader populations.

Response: We have added reference to this important work, thank you for your suggestion.

L149: In general the subheadings of the methods are not very well conceived, not well organized, and do not describe the content very well. They should be general but informative, and allow quick reference to the Results. The order should closely match that of the results. Please look at these carefully.

Response: The order of the methods chapters now matches the results section. We have also done our best to improve the subheadings in methods.

L152: state the year range

Response: The year range was 1992 – 2017, these details are mentioned in the revised manuscript.

L155: please report here, rather than in the results section, the details of the analyses carried out on the two NOAA grid cells close to the study site, and say that the estimate for these two cells is very similar to the estimate obtained for the entire range.

Response: Good suggestion, we now use only snowmelt near the field site for the analyses and justify the use of these data in the Methods section.

L164: please add here the mean percentage of body mass of the devices (including harness) for the tracked birds, and provide details on methods of device attachment to the birds.

Response: We used leg loop harness (the same as in Senner et al. 2015) weighing c.a. 1 g. Together with the 5g tag the attachment was about 1.2% of the body weight of females departing from the Wadden Sea and 1.5% of birds departing from Mauritania. These details have been added to this revision.

L181-183: very unclear. What do you mean? Do you have also 'median' dates? Medians of what distribution? Please clarify or rephrase.

Response: We removed this sentence as unnecessary.

L179: clutch initiation dates: for how many nests/year (give mean and min-max) has this variable been recorded? What was the protocol used? Standardized nest searches in a fixed area? Casual nest findings? Please clarify. How was this variable expressed? Median/mean first egg laying date in each year? Also, instead of 'clutch initiation date', use the more standard term 'first egg laying date'.

Response: We performed a standardized nest search effort at fixed plots with total area of several square km. But because godwits breed in very low densities, we report clutch initiation dates for all the 13 nests found in the region (see Supplementary Table 1 for all the nests found). We improved the explanation of methods.

L190: late-winter densities: how expressed? Mean values for all stations? This is not specified.

Response: Yes we used mean values of all the stations. Details are now specified.

L196: in the two areas: which ones? Clarify. Did you mean between the two areas?

Response: Exactly, we meant 'between the two areas' Clarified now.

L198-199: unclear how the lugworm densities in Balgzand and in Terschelling were expressed and compared. Mean of all sampling stations for each year at each site?

Response: Exactly! We have added these details.

L209-221: this part could be moved to the supplementary information as it is definitely too much technical for the normal reader. Simply explain the general purpose of the analyses and modelling here.

Response: We decided to keep this part in the Methods section as (1) the journal requires that 'Methods are contained within main paper wherever possible'; (2) it comes after the main part of the manuscript, so users can skip it and (2) we are still far below the word-count limitations of the Journal. However, to improve readability we have added a short introduction to the corresponding Methods section that allows nontechnical readers to skip the details. Note that reviewer 1 found the equations very important and actually asked us to add more details (e.g. details on the path analysis).

M L232: paragraph should be merged with the one at L243. In general, this part is very unclear. I am really getting lost here. You must clearly explain the rationale for computing refuelling rates from captures: which data were collected and for which purpose in a very schematic manner. Perhaps a flow chart might be of help. It is really difficult to figure out which kind of analyses have been made and for what purpose. If I understood correctly, you have calculated body mass at arrival for birds sampled with decoys, and refuelling body mass from birds captured at high tide roosts. From this you have computed annual refuelling rate, accounting for lugworm abundance. For each year, arrival date was estimated from citizen science data, and departure date was estimated as first arrival in Taimyr minus the mean number of days taken by satellite-tagged godwits to migrate from NL to Taimyr. The difference between departure and arrival for each year was the stopover duration. But (if I am not wrong) I could understand this only after looking at one of the supplementary figures 2/3! And all these steps are not clearly explained in the manuscript! I think that a representative example (one year) taken from this figures should be reported in the main text to illustrate the method used to derive each variable from which data.

Response: Thank you! Your suggestion to make an additional figure for a sample year makes a lot of sense. We incorporated it as new Figure 3. We have also done our best to improve wording in this section. Since both paragraphs were changed, we think it is not necessary to merge them anymore.

L232: the first sentence does not make sense, strictly speaking: capturing birds is not enough for estimating refuelling rates. Please rephrase. Also, the first trapping method is not explained: were mist-nets used? Only decoys and playback (to attract) are mentioned, but not the capture method.

Response: We rephrased the sentence and added details on capture techniques into the paragraph. In the Wadden Sea we used wind driven and pulled wilsternets, at Castricum a modified double clapnets for finches powered by elastic strings.

L239: add comma after 2016.

Response: Done.

L244: this is not clear. T0k was not estimated for each sex. If there is sex-specific migration, using a single value for both sexes may lead to erroneous conclusions. The reason for modeling refueling rates separately for each sex should be detailed.

Response: We do not have data on sex-specific differences in timing of migration (so arrival date and refuelling time did not depend on sex in our models), but because sexes in godwits differ in size and mass, we modelled arrival mass and refuelling rates as sex-dependent variables. We added this important detail into introduction, corresponding part of the Methods section and in the new figure (Figure 3).

L253: these values appear rather similar: again, is it really necessary to use sex-specific values?

Response: We preferred to keep sex-specific fuelling rates. We realized that we did not explain well how we estimated this parameter, and added a new figure (Figure 3) for clarification.

L257: all the description of how refuelling time was calculated should go in the previous paragraph, not here! This paragraph should only deal with the MARK analyses.

Response: Description of refuelling time estimation moved to the previous subchapter.

L271: 5.5 days: what is the variance around this figure? If it is large, it may not be appropriate to use this value for all data. On how many individuals was it computed?

Response: We tracked 8 migrations to Taimyr and they had mean duration of 5.16 days, SD = 2.09, median = 5.72. These details are now added to the results section.

L282: but in the case

Response: Corrected, thank you!

L290: break sentence before 'thus'

Response: Corrected, thank you!

Fig.1: add mean duration of the two segments of migration on the graphs (along tracks).

Response: Good idea, mean durations of the two migratory segments are now added to the figure.

Why 95% confidence bands are not shown for panel b? Either remove from all panels or add to all panels.

Response: We removed confidence intervals from the figure and incorporated the slope estimates and SEMs in Table 1, as suggested above.

L509: spring migratory tracks

Response: Done

L516: Legend to figure 2 is confusing and should be carefully revised to improve clarity and readability. Please make a further effort to improve it.

Response: We have completely reworked Figure 2 and the legend to make it clear.

L521-522: refuelling time in the y-axis label, but refuelling duration in the legend. Please be consistent and standardize terminology throughout!

Response: Corrected in the figure caption and also throughout the revised text.

L527: it is not only male survival which is in the supplementary, it is all the male-based analyses. Please clarify this in the legend.

Response: We see your point. We moved all male-related analysis results in the body of the manuscript.

REVIEWERS' COMMENTS:

Reviewer #1 (Remarks to the Author):

I have read the revised manuscript and think the authors addressed the earlier comments well. Just a few minor comments below:

1) Table 1, item 11): " There is not sex..." should be " There is no sex..."

2) Table 1: the delta AIC values are hard to interpret without knowing which of the two models had the lower AIC. Either give both AIC values, or give delta AIC but indicate which model had lower AIC, or always calculate AIC as 'simpler model' - 'more complex model' (so that positive values mean that the more complex model is better and negative values mean that the simpler model is better; or vice versa).

3) Table 1: some of these slope estimates are - I assume - on the logit scale. Make this clear. I understand that this table is already quite full but perhaps you can add this info into a table footnote. Or say "trend in logit annual survival" or similar.

4) P11 (without line numbers, I can't easily give you a more precise location...): " model was estimated" should be "model parameters were estimated"

5) Thank you for writing out the structural equation model explicitly. I think you could write them more concisely, however, by following the more usual notation. E.g. eq 1 could be written on a single line:

(Arrival Wadden Sea)_i = b_{1.1} X Time_i + e_i, e_i ~ N(0, sigma²_{arrWS})

And similar with the others.

Looking at your BUGS code, it seems to me that you used rather informative priors for the sigma parameters, i.e. a half-normal with precision 1. Are your results sensitive to this choice? It would be good to elaborate a bit on your choice of priors and also the fact that you apparently used the same starting values for the sigmas on all five chains. If there is no space for such detail in the methods section, at least add some text to the RMarkdown document. (Assuming that your results are not sensitive to these choices; if they are, you need to deal with this more carefully and argue for your choices in the ms itself.)

Reviewer #2 (Remarks to the Author):

The revised version of the ms by Rakhimberdiev and colleagues is considerably improved over the previous version. The authors have substantially revised the manuscript by adequately taking into account most of the reviewers' comments.

I still have a few minor edits and remarks that could be easily addressed in a revised version.

L36: show population declines

L38: We may expect

L43: ...hereafter godwits). Godwits are among the several...

L47: of ca. 5000 km, and spend

L48: next ca. 5000 km-long migratory

L59: sexually dimorphic, with...

L68: specify year range here

L69-75: I am really puzzled about this paragraph. I do not think it is in the right place! It should be moved as the first paragraph of the Discussion, summarizing results. I do not think it is a journal requirements, as other Nat Commun articles did not have such a paragraph.

L69: flies advanced

L73: shorter refuelling time was not

L89: the previously suggested

L92: migration paths

L93: migration path between

L97: initiation significantly advanced through time (-0.70....

L98: by birds showing a non-significant tendency to shorten...

L99: by 16% between 1995

L100: this sentence should be deleted. It is unclear before reading the subsequent paragraph.

L116: partly offset

L120: Tab. 1, statement 20).

L132-133: This sentence sounds unclear to me, and I suggest rephrasing it.

L171: please pay attention to caps and consistency: here you have used South-Eastern Taimyr, elsewhere, e.g. L191, south-east Taimyr, elsewhere simply Taimyr! Please check carefully and homogenize terminology.

L172: closest

L174: date on that at

L175: 1.00

L180: very unclear wording: maybe:for the two NOAA grid cells closest to the field site on the snowmelt dates...

L274 and 284: (1) and (2) are not defined, and should be removed.

L295: (estimated duration of.....data; see Results)

L314: report the exact number of resightings used in the analyses

L328: year (year)? Correct typo?

Figure 1: not sure if these soft colours are enough to link variables with data shown in a). Maybe consider more differentiated colours? Also note that here you report 5.2 days for the spring migration between Wadden sea and Taimyr, while in the text you report 5.5 days

L542: godwits equipped with ... 2016, together with the estimated duration of migration paths between Banc d'Arguin and Wadden Sea (3.8 d) and between Wadden Sea and Taimyr (5.5 d), and the...

L551: required in 2015 to maintain annual survival at the 1995 level has increased substantially

Figure 3: Add 'date' after 'Arrival...' in a)

L565: estimated spring migration time between Wadden Sea and Taimyr (5.5 days)...

L566: was estimated from godwits...

L568-569: unclear wording, consider the following: estimates with body mass values obtained from godwits refuelling in ... Also, at L568-569, be careful with use of caps (Arrival Mass, Arrival Date...), I would avoid it, here and elsewhere in the manuscript. Please check it carefully throughout for consistency in the use of caps.

**Best regards
Diego Rubolini**

REVIEWERS' COMMENTS:

Reviewer #1 (Remarks to the Author):

I have read the revised manuscript and think the authors addressed the earlier comments well. Just a few minor comments below:

1) Table 1, item 11): " There is not sex..." should be " There is no sex..."

Response: Corrected, thank you!

2) Table 1: the delta AIC values are hard to interpret without knowing which of the two models had the lower AIC. Either give both AIC values, or give delta AIC but indicate which model had lower AIC, or always calculate AIC as 'simpler model' - 'more complex model' (so that positive values mean that the more complex model is better and negative values mean that the simpler model is better; or vice versa).

Response: We have added the footnote that you suggested in which we explain that we calculated delta AIC as 'simpler model' - 'more complex model'.

3) Table 1: some of these slope estimates are - I assume - on the logit scale. Make this clear. I understand that this table is already quite full but perhaps you can add this info into a table footnote. Or say "trend in logit annual survival" or similar.

Response: We have added this clarification into the footnote, too.

4) P11 (without line numbers, I can't easily give you a more precise location...): "model was estimated" should be "model parameters were estimated"

Response: Corrected, thank you!

5) Thank you for writing out the structural equation model explicitly. I think you could write them more concisely, however, by following the more usual notation. E.g. eq 1 could be written on a single line:

$$(\text{Arrival Wadden Sea})_i = b_{1.1} \times \text{Time}_i + e_i, e_i \sim N(0, \sigma^2_{\text{arrWS}})$$

And similar with the others.

Response: We have changed all the structural equation model formulae to the suggested form.

Looking at your BUGS code, it seems to me that you used rather informative priors for the sigma parameters, i.e. a half-normal with precision 1. Are your results sensitive to this choice? It would be good to elaborate a bit on your choice of priors and also the fact that you apparently used the same starting values for the sigmas on all five chains. If there is no space for such detail in the methods section, at least add some text to the RMarkdown document. (Assuming that your results are not sensitive to these choices; if they are, you need to deal with this more carefully and argue for your choices in the ms itself.)

Response: Thank you for noticing! Although results were not significantly affected by choice of priors, we re-ran the model with uniform (from 0 to 10) priors and initial values for sigma parameters. We updated the markdown with model code, Supplementary Table 2 and Figure 1b with new parameter estimates.

Reviewer #2 (Remarks to the Author):

The revised version of the ms by Rakhimberdiev and colleagues is considerably improved over the previous version. The authors have substantially revised the manuscript by adequately taking into account most of the reviewers' comments.

I still have a few minor edits and remarks that could be easily addressed in a revised version.

L36: show population declines

Response: We would like to keep the sentence as it is as we already say “Migratory bird populations ... show declines”, so we do not consider adding ‘show population declines’ as improvement.

L38: We may expect

Response: Corrected, thank you!

L43: ...hereafter godwits). Godwits are among the several...

Response: We do not want to change “This population” to ‘Godwits’ here. Currently sentence says ‘This population is among the several long-distance migratory shorebirds that travel from wintering grounds in West Africa to...’. But not all the godwit populations travel from West Africa, just our population.

L47: of ca. 5000 km, and spend

Response: Corrected.

L48: next ca. 5000 km-long migratory

Response: Done, thank you!

L59: sexually dimorphic, with...

Response: Added comma.

L68: specify year range here

Response: corrected to ‘a 25-year period (1992-2016).’

L69-75: I am really puzzled about this paragraph. I do not think it is in the right place! It should be moved as the first paragraph of the Discussion, summarizing results. I do not think it is a journal requirements, as other Nat Commun articles did not have such a paragraph.

Response: We agree that this paragraph is a bit unusual but it is required by the journal.

L69: flies advanced

Response: Corrected, but also changed to present tense, as requested by the editor.

L73: shorter refuelling time was not

Response: Done.

L89: the previously suggested

Response: Corrected.

L92: migration paths

Response: we prefer having migration legs but not paths here, as were focus not on the paths but on the segments of migration in general.

L93: migration path between

Response: please see the previous reply.

L97: initiation significantly advanced through time (-0.70....

Response: Significance of the changes is presented immediately after the statement, so we prefer to avoid unnecessary word 'significant' here.

L98: by birds showing a non-significant tendency to shorten...

Response: Non-significant P value of the trend is presented immediately after the statement, so we prefer to not add unnecessary word 'non-significant' here.

L99: by 16% between 1995

Response: Corrected, thank you.

L100: this sentence should be deleted. It is unclear before reading the subsequent paragraph.

Response: We deleted this sentence.

L116: partly offset

Response: Corrected.

L120: Tab. 1, statement 20).

Response: Following editor's advice we decided on not abbreviating rows in table one and are now using "Table 1, row 20").

L132-133: This sentence sounds unclear to me, and I suggest rephrasing it.

We seem to be in disagreement here, and like to maintain what for us is a clear sentence (Rather than the use of multiple sites simply being a liability, it may provide opportunities for among-season compensation.), as it was.

L171: please pay attention to caps and consistency: here you have used South-Eastern Taimyr, elsewhere, e.g. L191, south-east Taimyr, elsewhere simply Taimyr! Please check carefully and homogenize terminology.

Response: Thank you for noticing! We now use South-Eastern Taimyr in all the cases.

L172: closest

Response: Corrected, thank you.

L174: date on that at

Response: Great suggestion, thank you.

L175: 1.00

Response: Done.

L180: very unclear wording: maybe:for the two NOAA grid cells closest to the field site on the snowmelt dates...

Response: Agreed, thank you for the great suggestion.

L274 and 284: (1) and (2) are not defined, and should be removed.

Response: We removed these unnecessary numbers.

L295: (estimated duration of.....data; see Results)

Response: good suggestion, thank you!

L314: report the exact number of resightings used in the analyses

Response: we now report the total of 4813 resightings.

L328: year (year)? Correct typo?

Response: thank you for noticing, corrected now.

Figure 1: not sure if these soft colours are enough to link variables with data shown in a). Maybe consider more differentiated colours?

Response: We prefer to keep the current colour scheme.

Also note that here you report 5.2 days for the spring migration between Wadden sea and Taimyr, while in the text you report 5.5 days

Response: Corrected, thank you!

L542: godwits equipped with ... 2016, together with the estimated duration of migration paths between Banc d'Arguin and Wadden Sea (3.8 d) and between Wadden Sea and Taimyr (5.5 d), and the...

Response: Great suggestion, thank you!

L551: required in 2015 to maintain annual survival at the 1995 level has increased substantially

Response: Changed wording as suggested, thank you!

Figure 3: Add 'date' after 'Arrival...' in a)

Response: Done.

L565: estimated spring migration time between Wadden Sea and Taimyr (5.5 days)...

Response: Done.

L566: was estimated from godwits...

Response: Corrected, thank you.

L568-569: unclear wording, consider the following: estimates with body mass values obtained from godwits refuelling in ...

Response: Good suggestion, thank you!

Also, at L568-569, be careful with use of caps (Arrival Mass, Arrival Date...), I would avoid it, here and elsewhere in the manuscript. Please check it carefully throughout for consistency in the use of caps.

Response: We have removed all caps in the variable names.

Best regards
Diego Rubolini